# Therapeutic Advancements in Metal and Metal Oxide Nanoparticle-Based Radiosensitization for Head and Neck Cancer Therapy

**DOI:** 10.3390/cancers14030514

**Published:** 2022-01-20

**Authors:** Poornima Dubey, Mathieu Sertorio, Vinita Takiar

**Affiliations:** 1Department of Radiation Oncology, University of Cincinnati Barrett Cancer Center, 234 Goodman Street, ML 0757, Cincinnati, OH 45267, USA; dubeypa@ucmail.uc.edu (P.D.); sertormu@ucmail.uc.edu (M.S.); 2Cincinnati Department of Veterans Affairs (VA) Medical Center, 3200 Vine St., Cincinnati, OH 45220, USA

**Keywords:** head and neck cancer (HNC), radiosensitizer, metal nanoparticle (MNP), metal oxide nanoparticle, clinical translation, nanoparticles, radiation, protons, photons

## Abstract

**Simple Summary:**

Though radiation therapy remains a primary modality for head and neck cancer (HNC) management, the collateral damage to normal surrounding tissues and tumor relapse represent major challenges. Hence, it is imperative to develop safer and more effective HNC therapies. Metal and metal oxide nanoparticle-based radiosensitizers have been explored for their potential to overcome these challenges. The key impetus of this review was to shed light on ongoing metal and metal oxide nanoparticle-based radiosensitizers’ development and to address their success in in vitro and in vivo HNC models and in clinical translation.

**Abstract:**

Although radiation therapy (RT) is one of the mainstays of head and neck cancer (HNC) treatment, innovative approaches are needed to further improve treatment outcomes. A significant challenge has been to design delivery strategies that focus high doses of radiation on the tumor tissue while minimizing damage to surrounding structures. In the last decade, there has been increasing interest in harnessing high atomic number materials (Z-elements) as nanoparticle radiosensitizers that can also be specifically directed to the tumor bed. Metallic nanoparticles typically display chemical inertness in cellular and subcellular systems but serve as significant radioenhancers for synergistic tumor cell killing in the presence of ionizing radiation. In this review, we discuss the current research and therapeutic efficacy of metal nanoparticle (MNP)-based radiosensitizers, specifically in the treatment of HNC with an emphasis on gold- (AuNPs), gadolinium- (AGdIX), and silver- (Ag) based nanoparticles together with the metallic oxide-based hafnium (Hf), zinc (ZnO) and iron (SPION) nanoparticles. Both in vitro and in vivo systems for different ionizing radiations including photons and protons were reviewed. Finally, the current status of preclinical and clinical studies using MNP-enhanced radiation therapy is discussed.

## 1. Introduction

Head and neck cancer (HNC) is the sixth most common cancer worldwide, with an estimated incidence of 53,260 cases and 10,750 deaths in 2020 alone [1]. HNC originates from the epithelial mucosal layer of the upper aerodigestive tract and includes tumors of the nasopharynx, oral cavity, pharynx, and larynx. The chief risk factors for HNC have historically been tobacco use and alcohol consumption, but human papilloma virus (HPV) infection is now a predominant predisposing cause [2,3]. As with any malignancy, HNC treatment depends on the pathological and clinical stage of the tumor. Unfortunately, HNC has comparatively low survival rates compared to many other solid tumors, in part because an estimated 60% of patients are diagnosed with advanced disease (stage III and IV). In locally advanced HNC cases, post-operative radiation and definitive chemoradiation (CRT) are commonly used treatment paradigms [4].

Radiation treatment involves the direct deposition of select doses of energy to target tissues [5,6,7]. Ionizing radiation (in the form of X-rays, γ-rays, electrons, neutrons, and charged particles), such as that used in radiation therapy (RT), can damage cells either by direct interaction with critical targets or indirectly through free radical generation [8,9]. The main limitation of RT is the absence of spatial control on the deposition of energy, i.e., untargeted dose deposition occurs in normal neighboring tissues, causing unwanted damage and limiting the delivery of an optimal energy dose to the malignant tumor. Recent advancements in radiation delivery have resulted in increased precision. As precision has increased, so have attempts to improve tumor control by increasing radiation dose. While tumor cell damage is directly proportional to the energy of the dose given, increasing the radiation dose has proven insufficient to advance the tumor control probability (TCP) for numerous radioresistant HNCs [5,6,7,10,11].

To improve therapeutic outcomes, additional strategies must be developed that maximize tumor control while minimizing surrounding tissue damage. This is particularly imperative in HNC because the primary tumor is often situated adjacent to radiosensitive organs with critical functions such as the salivary glands, thyroid, larynx, and swallowing apparatus [12]. To address these challenges, during the last few decades there has been increasing interest in exploiting various nanomaterials (size 10–1000 nm) for drug delivery. These materials include nanoliposomes, nanomicelles, nanofibers, drug nanoparticle conjugates, and nanocomposites. Along with metal nanoparticles (MNP), these novel approaches allow for both complex cargo delivery and cancer theranostic (therapy and diagnosis) applications [13,14,15,16,17,18,19,20,21,22,23]. Among them, high atomic number metal-based NPs have attracted wide attention in the field of radiotherapy. MNPs are taken up by cells depending on factors such as size, surface, density, and shape. MNPs passively diffuse into the tumor tissue by a phenomenon referred to as the enhanced permeability and retention effect (EPR) [24]. Upon absorption or deposition of a high-energy dose of ionizing radiation, MNPs generate a cascade of Auger electron emissions within the tumor tissue. Therefore, in the presence of MNPs there is an increase in the total number of Auger electrons within the tumor for the same amount of radiation, resulting in increased cellular damage [25,26]. Thus, MNPs and metal oxides-based nanoparticles are appealing for their potential to function as radiosensitizers [25,27] or chemical agents specifically designed to sensitize tumor tissues to RT [27,28,29,30,31]. Here, we summarize the current research using metal and metal oxide-based nanoparticle radiosensitizers with a specific focus on the recent progress and development of metal nanoparticles as radiosensitizers for HNC treatment.

## 2. Metal Nanoparticles (MNPs) Function as Radiosensitizers

The ability of high atomic number (high Z) MNPs to augment radiation dose localization within the tumor while minimizing collateral normal tissue damage has the potential to transform RT [27,28,31]. Nanoparticles induce reactive oxygen species (ROS) through interaction with incoming proton/photons; hydroxyl ions (OH−) and free radicals are the main products of water radiolysis by irradiation. Secondary electrons further augment radiation dose locally within the tumor and mediate ROS generation either by charge transfer to produce O2− from dissolved oxygen molecules or by energy transfer from fluorescent X-rays or bremsstrahlung X-rays. The probability of this interaction depends on the incident radiation energy and the atomic Z value of the particle’s atoms [31,32]. MNPs amplify the effect of ionizing radiation by increasing the locally absorbed radiation dose. This increased radiation dose results in increased ROS-led oxidative stress, which leads to increased DNA double-strand breaks (DSB) along with other subcellular damage and ultimately increased cell death (Figure 1, Figure 2 and Figure 3). It has been well documented that irradiated nanoparticles, preferentially sequestered by tumor cells due to diffusion and the EPR effect, emit showers of secondary electrons that consequently increase water radiolysis around the sites of nanoparticle accumulation and damage important biomolecules, including DNA, protein, and lipids [33,34]. Notably, it has been documented that while DNA is located in the cell nucleus, the nucleus is inaccessible even for nanoparticles of ultrafine dimensions such as gadolinium (Gd)-based nanoparticles (~2.5–10 nm) [35,36]. However, some nanoparticles concentrate around the cell nucleus or are specifically directed to the endoplasmic vesicles and reticulum; therefore, some secondary electrons inevitably reach and damage the chromatin [36]. Hence, for the optimal design of MNPs, their physiochemical properties of shape, size, zeta potential (stability in the physiological medium), EPR effect, immunological response, and biocompatibility are of paramount importance [37].

### 2.1. Proton-Based Radiosensitization by MNPs

The magnitude and localization of MNP-based radiosensitization vary depending on the type and energy of the ionizing radiation used as well as the characteristics of the nanoparticles including shape, size, surface coating, and concentrations. Over the last decade, the number of facilities offering proton radiation has continued to increase as has the clinical usage of proton therapy for cancer treatment [38]. This is mainly due to the potential for reduced side effects with comparatively lower integral dose than conventional X-ray radiotherapy [38,39,40]. The vast majority of proton energy dose gets deposited at the end of the proton range (Bragg Peak), in a targeted tumor volume minimizing dose to normal tissue along the beam path. Notably, the dosimetric advantages of proton radiation can be enhanced by increasing the energy deposition in the target with the addition of high Z MNPs in the target tumor tissue [41,42,43]. The benefit of nanoparticles in combination with photon radiation is chiefly due to the intensification in photoelectric interactions that is present at low energy. However, proton radiation yields dense ionization with high linear energy transfer (LET). LET refers to the amount of energy transferred from the material from an ionizing particle per unit distance, and a high LET dose is attributed to particles with substantial mass and charge including alpha particles, which attenuate radiation more quickly, delivering a relatively higher dose over a smaller distance [38,39,40,41,42,43,44]. Radiosensitization or dose augmentation produced by the addition of high Z atom MNPs in target tumor tissues with photon radiation has been widely examined and recognized in the literature, while the use of MNPs with proton beam radiation continues to be an active area of research [45,46].

### 2.2. Gold Nanoparticle Applications

Gold nanoparticles (AuNPs), where Au is (Z = 79), with their relatively large number of protons within their nucleus, have numerous advantageous features including easy surface functionalization capacity for targeted delivery, biocompatibility, and a relatively strong photoelectric absorption coefficient compared to soft tissues. The enhanced photoelectric effect, whereby the electrons are ejected from the innermost atomic orbital, then creates a vacancy that is filled by electrons from the neighboring outermost shell jumping, which leads to a cascade of electrons known as the Auger electron cascade. This process leads to the generation of low-energy electrons within short nanometer or micrometer distances, leading to a significant ionization process, which can cause DNA damage as discussed above [43,46,47]. 

In the therapy of solid tumor, AuNPs have been widely explored for their applications in solid tumor therapy including (1) Photothermal therapy (PTT), by using electromagnetic radiation in the treatment of various medical applications; (2) Photodynamic therapy (PDT), by using light-sensitive medicine upon light irradiation for tumor therapy; (3) nano-based chemotherapy; (4) radiofrequency-mediated hyperthermia (HT), in which body tissue is exposed to high temperatures to destroy tumor cells or to sensitize the tumor to the effects of radiation and certain chemotherapeutic agents; and (5) gold nanoparticle (AuNP, Nanogold)-enhanced radiation therapy (NRT) [47,48,49]. In 2000, Herold et al. reported that, upon kilovoltage X-rays’ exposure, the gold microspheres could produce biologically effective dose enhancement [50]. In 2010, Hainfield et al. demonstrated the effect of AuNPs in improving radiation therapy and their efficacy for the treatment of a radiation-resistant and aggressive HNC mouse model, subcutaneous (sc) SCCVII [51]. The subcutaneous (sc) SCCVII tumor-bearing mice were irradiated with X-rays with and without previous intravenous administration (iv) of AuNPs and then analyzed by temporal fractionation, radiation dose, beam energy, and HT. It was observed that AuNPs were more efficient at 42 Gy than at 30 Gy (energy was kept similar at 68 keV) compared to the samples without AuNPs. Hence, it was established that HT and RT functioned synergistically and AuNPs augmented this synergy, reducing the TCD50 (tumor control dose 50%) and increasing the long-term survival. Though the underlying mechanisms for these differences were not obvious, this study clearly demonstrated that AuNPs augment the effect of radiation on a radioresistant mouse HNC and that radiation dose, energy, and HT together affect their efficacy, suggesting the possibility of using AuNPs as radiosensitizers for improved HNC therapy [51]. 

Koonce and colleagues designed and evaluated the effect of PEGylated AuNPs and tumor necrosis factor-α (TNF) as one nano-entity named CYT-6091 (CytImmune, http://www.cytimmune.com/) (accessed on 12 October 2021). When CYT-6091 was combined with X-rays in vivo, head and neck tumor growth was inhibited. Because CYT-6091 has already passed phase 1 trials (NCT00356980 and NCT00436410), this combination is ripe for clinical translation [52]. In another one of the first studies examining NP HNC, Teraoka et al. explored the HSC-3 (human tongue squamous cell carcinoma) cells to demonstrate that 4-Gy X-ray irradiation could significantly reduce total cell count, and the addition of 1.0-nM AuNPs increased cell death even further. Notably, they found that the reduction of total cell number by 4-Gy X-ray irradiation alone and when combined with 1.0-nM AuNPs was attributed to the induction of apoptosis [53].

Gold nanoparticles may be particularly useful for overcoming tumor resistance to radiotherapy, a major challenge in the therapy of HNCs, accounting for a 40% local failure despite aggressive radiation treatment. Interestingly, several studies have been undertaken to study the role of AuNPs in combination with chemoradiotherapy. The monoclonal antibody Cetuximab, an epidermal growth factor receptor inhibitor (EI), is a targeted molecular therapy used in combination with radiotherapy for HNCs. Popovtzer et al. in 2016 attempted to develop a technique that would overcome tumor radioresistance by using cetuximab-targeted gold nanoparticles (AuNPs), using a clinically relevant 6-MV energy beam delivered as a single 25-Gy radiation fraction. This study demonstrated that targeted AuNPs enhanced the radiation effect and had a significant impact on tumor growth. This study explored the biological mechanisms of radiation enhancement by AuNP, which were corroborated with earlier and enhanced apoptosis and decreased CD34 levels, confirming reduced vascularization and potentially angiogenesis inhibition. Further, PCNA staining confirmed diminished DNA repair mechanisms after AuNPs, suggesting a radiosensitizing effect [54]. Importantly, AuNPs by themselves have proven to be safe with no evidence of toxicity in mice and no additional toxicity when combined with Cetuximab. Taken together, this study evaluated the effectiveness of AuNP in combination with Cetuximab, suggesting safety and a potential benefit to combining this targeted therapy with AuNP.

AuNPs have also been incorporated into multifunctional nanoplatforms. For instance, Davidi at al. designed a single nanoplatform, consisting of AuNPs surface coated with cisplatin and glucose (referred to as CG-AuNPs). The designed CG-AuNPs’ nanoentity would simultaneously act as a radiosensitizer (due to AuNP), along with serving as a drug delivery agent, by specifically delivering the chemotherapeutic agent cisplatin to the tumor cells, and as an effective CT contrast agent. This study demonstrated that the multifunctional CG-AuNP effectively penetrated into the HNC in vivo model and demonstrated a similar cytotoxic effect as cisplatin. Furthermore, their study demonstrated improved tumor growth inhibition in vivo by CG-AuNPs in combination with RT, in comparison to RT and to RT + cisplatin combination. Additionally, they demonstrated the feasibility of using CG-AuNPs as a CT contrast agent. This study offers a unique multi-functional nanoplatform for HNC therapy. In order to understand the effectiveness of this nanoplatform for prevalent clinical practices, further studies should be undertaken with multiple CG-AuNPs’ injections along with multiple radiation sessions [55].

In 2020, Kashin et al. developed a nanoplatform by combining the AuNPs with the tyrosine kinase inhibitor, AG1478, an epidermal growth factor receptor inhibitor (EI), for enhanced radiation effects on HNCs. The conjugation of AG1478 on AuNP surfaces was confirmed by Surface-enhanced Raman scattering (SERS), by measuring an adsorption equilibrium of AG1478 to AuNPs. Cellular uptake studies done with transmission electron microscopy (TEM) demonstrated similar cellular internalization rates for the AuNP alone and AuNP with AG1478. Comparisons in cell numbers, proliferation, cell death, and cell migration with or without 60-nm cAuNP (1.0 nM), AG1478 (0.5 μM), and irradiation (4 Gy) were done between control and treated groups. They observed that the combination of AuNP and AG1478 inhibited cell proliferation more than AG1478 alone, whereas the combination of RT, AuNP, and AG1478 significantly decreased the total cell numbers in comparison with the combination of RT and of AuNP only. The combination of AuNP and AG1478 and irradiation induced more apoptosis than AG1478 and irradiation [56]. While tyrosine kinase inhibitors are not in common use for HNC therapy, this study suggested that combination treatment with AuNPs may offer enhanced efficacy.

Cancer theranostics represents the intersection of therapy and tumor bioimaging. In 2019, Jia et al. synthesized gold nanoclusters as small-sized radiosensitizers with strong adsorption, scattering, and emission properties. The developed radiosensitizer was comprised of a well-defined, gold-levonorgestrel nanocluster encompassing Au8(C21H27O2)8 (Au8NC) with bright luminescence (58.7% quantum yield) and reasonable biocompatibility (Figure 4). Tumor size decreased after one radiotherapy treatment with the Au8NCs in vivo (Figure 5). It was observed that the radioenhancement occurred in a ROS mediated-manner, causing cell apoptosis. The nanosensitizer not only reduced the X-ray dose but was also reported to decrease the side effects of radiation in normal tissues [57,58,59].

### 2.3. Gadolinium-Based Nanoparticles

In addition to the more widely explored AuNPs, gadolinium-based nanoparticles, also known as AGuIX (NHTheraguix, Crolles, France), have been explored for their role as radiosensitizers for HNCs. AGuIX are composed of a polysiloxane matrix decorated with a chelating species of 1-, 4-, 7-, and 10-tetraazacyclododecane-1 and 4-, 7-, and 10-tetraacetic acid (DOTA) at their surface where DOTA ligands chelate with the Gd3 + ions, allowing their application as MRI contrast agent. DOTA also allows for detection by positron emission tomography (PET) and single-photon emission computed tomography (SPECT). AGuIX have been explored for their radiosensitizing properties in vivo and in vitro, owing to their high atomic number (Z, 64). Rapid blood and renal clearance are achieved due to their small size (<6 nm). These particles also significantly accumulate within the tumors by passive targeting via the EPR effect. Initially, the enhancement of radiotherapy by AGuIX was demonstrated in an HNC-based orthotropic animal model [60,61,62,63,64]. For this study, orthotopic HNC tumors were generated by injecting tumor fragments of human head and neck CAL33-Luc or subcutaneous tumor SQ20B into the tongue of mice. A single, 10-Gy dose of RT was delivered after incomplete resection. Interestingly, AGuIX nanoparticles accumulated in the tumor cells remaining after surgery. The study demonstrated statistically significant improvement in radiotherapy efficiency with AGuIX. Notably, all of the irradiated groups (including irradiated only and irradiated 1 h post IV of AGuIX) showed a substantially slower tumor growth compared to the control group. In order to optimize the clinical utility of AGuIX, there is a need to evaluate additional radiation protocols, either by altering the radiation dose or by changing the number of injected nanoparticles, by adjusting the time between the injection of nanoparticles and the radiation dose, or by exploring fractionated irradiation protocols [60,61,62,63,64]. In summary, in the very first trial, the 

Gd-based AGuIX nanoparticles demonstrated the enhancement of radiotherapy in an HNC orthotropic animal model. 

### 2.4. Miscellaneous MNPs Explored as Radiosensitizer for HNCs

In addition to the nanoparticles discussed above, in a recent study, Yu at al., developed a novel nanocomposite, Ag/C225, comprised of silver nanoparticles (AgNP) tailored with an epidermal growth factor receptor specific antibody (C225). AgNP and Ag/C225 inhibited the proliferation of nasopharyngeal carcinoma epithelial (CNE) cells in a time- and concentration-dependent manner. Interestingly, flow cytometry revealed that AgNP and Ag/C225 induced the apoptosis of CNEs and abrogated G2 arrest; the latter effect was more marked with Ag/C225 than with AgNPs only. Colony forming assay showed that AgNPs and Ag/C225 augment the radiation sensitivity of CNEs. Combining X-ray irradiation with either AgNPs or Ag/C225 decreased the expression levels of DNA damage and DNA repair proteins Ku-70, Ku 80, and Rad51 by Western blotting. The Ag/C225 was also found to be more efficient in tumor killing than AgNPs only. These preliminary results suggest that AgNP-based agents may be considered for use as radiosensitizers during the treatment of human nasopharyngeal carcinoma, and this is significant as radiation dose to this area of the head and neck is often limited by proximity to the brain [65].

## 3. Metal Oxide Nanoparticles as Radioenhancers for HNCs

Along with MNPs, metal oxide nanoparticles are also widely explored nanomaterials for their ability to radiosensitize. Iron oxide nanoparticles (SPIONS) are metal oxide nanoparticles that offer many exciting features that can be exploited in MRI imaging, in creating drug delivery systems, and in enhancing cancer therapeutics [66,67,68,69]. In a recent study, Thapa at al. designed hyaluronan-mediated dextran-coated superparamagnetic iron oxide nanoparticles (HA-DESPIONs), which retained the native biological recognition of HA receptor CD44, which binds to hyaluronan. CD44 is a marker of cancer stem cells (CSCs) in HNC and is a promising target for a variety of anticancer therapeutic approaches [66]. HA-DESPIONs-mediated cytotoxicity, radiosensitivity, and hyperthermia were evaluated in CD44-expressing HNC cell lines using clinically relevant radiation beam energies and temperatures. Radiosensitizing properties and hyperthermia-induced toxicity of HA-DESPIONs were evaluated in both flow-based CD44 sorted and CD44 unsorted cells in combination with 2-Gy photon and at 40 °C, 41 °C, and 42 °C using clonogenic assays. The studies suggested that HA-DESPIONs were nontoxic at moderate concentrations; however, interestingly, they did not radiosensitize the cell lines directly. Moreover, there was no difference in the radiosensitivity of CD44 high and CD44 low cells. Nevertheless, HA-DESPIONs improved the effect of hyperthermia, which, in turn, caused reduced cell survival driven by an increase in apoptosis. This study puts forward scope for utilization of chemotherapeutic payload-attached HA-DESPIONs in amalgamation with radiation for specific CD44-mediated targeting of CSCs [66]. Alongside SPIONS, a few other metal oxide nanoparticles have also been explored for their radiosensitization efficacy.

Zinc oxide nanoparticles (ZnO-NPs) are also metal oxide nanoparticles that have been shown to induce photocatalytic cell death in HNC cell lines in vitro. The antitumor response against HNC in vitro has been shown to be related to autophagy-mediated cell death. Furthermore, photo-stimulated ZnO-NPs have been established to synergistically improve the cytotoxic effects of chemotherapeutic agents such as paclitaxel and cisplatin against HNC cell lines. As noted above, the efficacy of Cetuximab in the treatment of HNC, alone or in combination with other cytostatic drugs, has been established intensively [70]. To enhance this efficacy further, Gehrke et al. evaluated the effect of ZnO-NPs on the antitumor properties of Cetuximab in HNC in vitro. They selected two HNC cell lines (FaDu and HLaC-78) and evaluated cytotoxicity in a concentration- and time-dependent manner by treating them with 0.1, 1, or 10 mM Cetuximab as well as 0, 0.1, or 1 mg/mL ZnO-NP after 24, 48, and 72 h of incubation with Cetuximab and ZnO-NPs. Counterintuitively, the study showed that ZnO-NPs antagonized the anti-tumor properties of Cetuximab in a time- and dose-dependent manner. These findings shed light on an inhibitory interaction of ZnO-NPs with Cetuximab, which warrants additional research [70]. 

## 4. MNPs-Based Radiosensitizers in Clinical Trial 

In spite of the potential of NPs to induce radiosensitization in tumor cells, there are numerous challenges to clinical translation [71,72,73,74]. Therefore, there have only been a limited number of clinical trials involving NPs in HNC, largely liposome-based and unrelated to improving radiation efficacy [75,76,77]. These challenges include discrepancies found in the mechanisms of action of NPs, few studies evaluating long-term side effects, and the inadequate demonstration of therapeutic efficiency in radiation dose delivery in megavoltage (MeV) energies, at which radiotherapy is clinically performed for head and neck cancer [58,59]. Additionally, the optimal radiation dose to be used with NPs is unclear as NPs upsurge the dose deposition greatly within their locality and this could cause increased exposure to organs at risk [78]. Additionally, NPs may alter cellular communications, potentially affecting clinical outcomes. Therefore, additional studies are required before clinical translation. Nonetheless, using NPs can be a strength not just to radiosensitize cells but also to provide contrast as they can be imaged. Although beyond the scope of the present review, NPs can serve as theranostic agents, combining therapeutic and imaging potential into a single NP, thus improving precision and outcomes of treatment delivery. Regardless of the possible applications, without understanding the mechanisms behind the biological effects within the tumor cells, it is difficult to robustly move towards clinical applications [13,73,76,79]. Nonetheless, there are a few notable clinical trials that have opened in the recent past, furthering the development of metal and metal oxide nanoparticle-based radiosensitizers for HNCs among other solid tumors. These trials are listed in Table 1. Among them, only hafnium oxide is being actively evaluated in a currently enrolling phase1/2 clinical trial.

Hafnium nanoparticles represent a new class of materials with high electron density, in the form of crystalline 50-nm nanoparticles (HfO2-NP). This high electron density augments radiation absorption, thereby intensifying the radiation dose deposited. These “hot spots” of energy deposition, within the tumor cells, result in more focused and efficient cell killing. Preclinical studies have established augmented cancer cell death in vitro and noticeable antitumor efficacy in vivo due to HfO2-NP in combination with irradiation, over RT alone. HfO2-NP (NBTXR3), administered as a single intertumoral injection and activated by RT, is currently being evaluated in a phase 2 clinical trial for head and neck cancer [NCT04862455]. So far, patients treated in phase 2 demonstrate good local and systemic tolerance to the product up to the highest dose level with RT delivered as planned, confirming safety. As the furthest along in clinical evaluation, NBTXR3 nanoparticles constitute a rising hope for head and neck cancer patients that could lead to a decrease in the long-term adverse properties of RT and an improved quality of life, associated with strong locoregional control. NBTXR3 in combination with RT is also being evaluated in clinical trials for soft tissue sarcoma, prostate, liver, and rectal cancer, with a promising risk–benefit ratio assessment and efficacy, signifying the potential to improve treatment outcomes for head and neck cancer [80].

## 5. Conclusions and Future Directions

HNC is a heterogeneous and complicated disease with significant mortality and treatment-related morbidity. The therapeutic success of advanced HNC treatment, which is often based on ionizing radiation therapy with or without chemotherapy, has been limited by severe long-term side effects and radioresistance. The challenge, as emphasized in this review, is how to widen the therapeutic ratio so as to effectively target the radiation to the tumor while preserving nearby tissues. One promising approach is to selectively pre-sensitize tumor cells with metallic nanoparticles prior to delivering highly focused RT. As nanoparticles increase the efficacy of the delivered radiation, increased tumor cell death is anticipated (Table 2). Physiochemical properties of metal and metal oxide-based nanoparticles (including material, composition, size, shape, surface functionalization, stability, etc.), tumor cell-type-specific response, and experimental conditions (including type of radiation, radiation doses, nanoparticle concentrations, incubation times, etc.) could be diverse, presenting both opportunities and challenges. These complexities have limited the translation of laboratory findings in HNC cell lines to the clinic for HNC patients. Prerequisites to further advancement include resolving discrepancies between in vitro and in vivo data and developing more clinically relevant models such as HNCs’ orthotopic xenografts. Though the “physics” behind nanoparticle-mediated radiosensitization has been well explained, there remain additional opportunities to explore the biological applications in medicine.

## Figures and Tables

**Figure 1 cancers-14-00514-f001:**
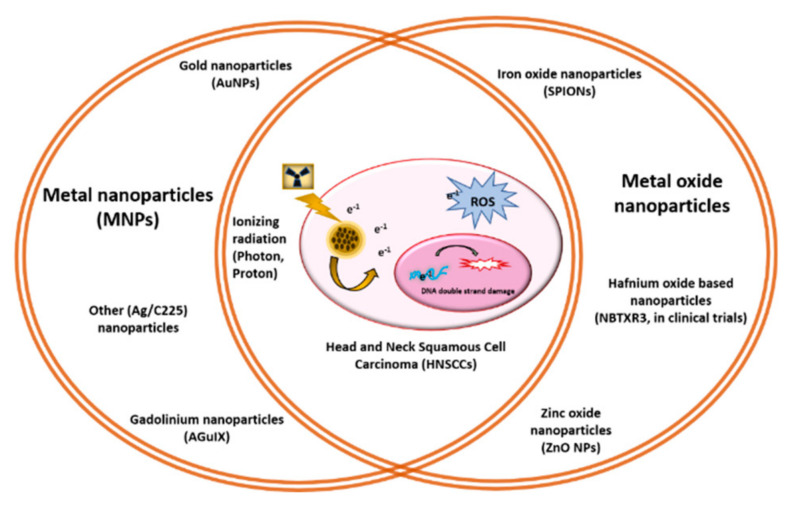
Schematic of metal and metal oxide nanoparticles.

**Figure 2 cancers-14-00514-f002:**
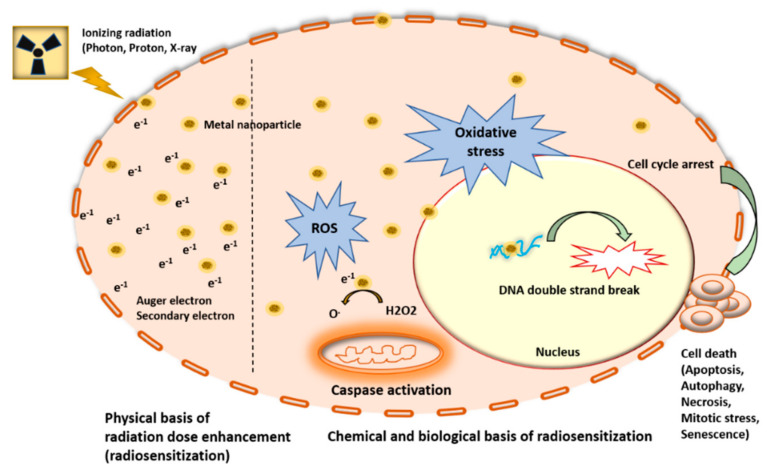
Schematic representation of physical bases (interaction of radiation with nanoparticle leading to generation of Auger electron cascade), chemical basis (generation of ROS due to enhanced radiolysis of water), and biological basis (DNA damage eventually leading to cell death) of metal nanoparticle-based radiosensitization in HNCs.

**Figure 3 cancers-14-00514-f003:**
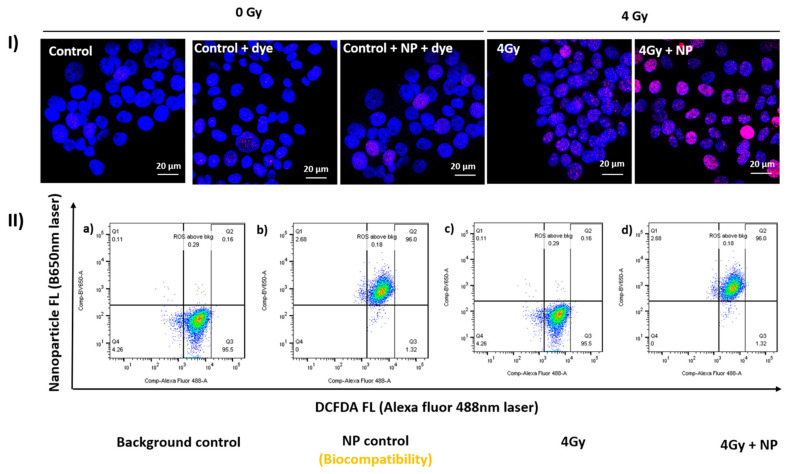
Enhanced DNA damage and ROS generation with radiation in the presence of gold nanoparticles. (**I**) The yH2AX-based double-stranded DNA damage studies performed in human tongue carcinoma Cal27 cell lines demonstrated increased yH2AX foci with gold nanoparticles in combination with a single 4-Gy photon dose. (**II a–d**) The dot plot from left to right illustrates flow cytometry-based quantitation of 2′,7′-Dichlorofluorescin (DCF) green FL emitting dye under 488-nm laser channel with increased ROS production after radiation treatment (Cesium-137 source) in the presence of a gold nanoparticle upon 4-Gy irradiation. (Unpublished data, Takiar laboratory, University of Cincinnati).

**Figure 4 cancers-14-00514-f004:**
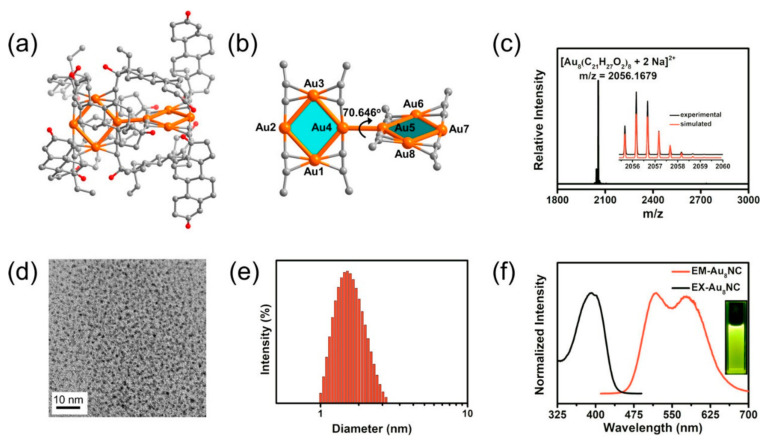
Structure and characterization of the Au8NCs. Perspective views of the Au8NCs showing (**a**) the molecular structure and (**b**) the dihedral angle formed by the planes of two tetranuclear units. Color codes: orange indicates Au, red indicates O, and gray indicates C. Hydrogen atoms and some carbon atoms were omitted for clarity. (**c**) Positive mode ESI-TOF-MS spectrum of the Au8NCs. The inset shows an enlarged portion of the spectrum, showing the measured (black line) and calculated (red line) isotopic distribution patterns. (**d**) TEM image of the Au8NCs. (**e**) DLS analysis of the Au8NCs. (**f**) Normalized excitation and emission spectra of the Au8NCs (inset: image of Au8NCs under 365-nm laser excitation in phosphate buffer, 10 μM) at room temperature. This work is reprinted from Atomically Precise Gold–Levonorgestrel Nanocluster as a Radiosensitizer for Enhanced Cancer Therapy, published by Tong-Tong Jia, Guang Yang, Sai-Jun Mo, et al. with copyright© 2019, American Chemical Society.

**Figure 5 cancers-14-00514-f005:**
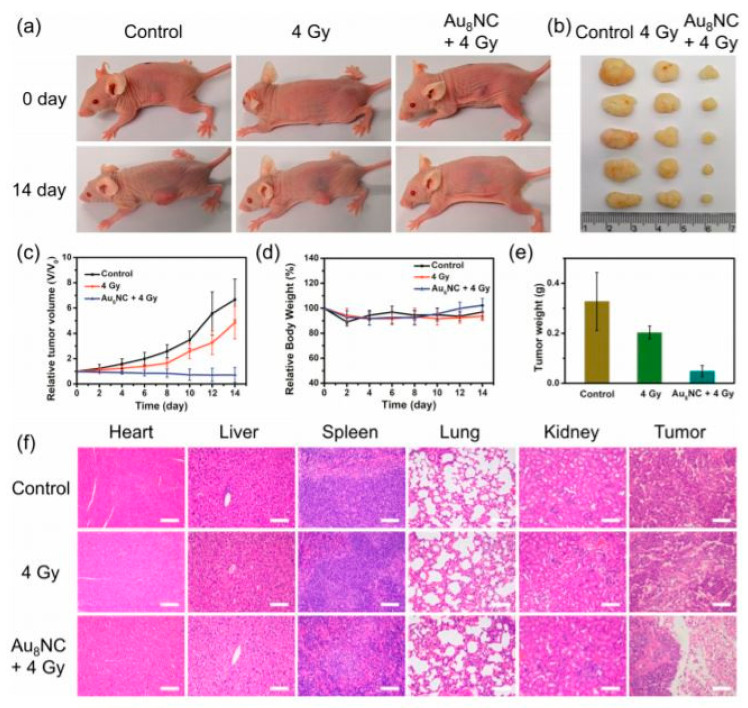
In vivo tumorigenicity assay of Au8NCs under different conditions. (**a**) Representative images of mice under various conditions at days 0 and 14. (**b**) Images of dissected tumors. (**c**) Relative tumor volume curves of the mice. (**d**) Relative mouse body growth curves. (**e**) Statistical results of the tumor weights. (**f**) H&E histological staining of excised organs and tumor slices. Scale bar: 100 μm. This work is reprinted from Atomically Precise Gold–Levonorgestrel Nanocluster as a Radiosensitizer for Enhanced Cancer Therapy, published by Tong-Tong Jia, Guang Yang, Sai-Jun Mo, et al. with copyright© 2019, American Chemical Society.

**Table 1 cancers-14-00514-t001:** Metal oxide-based nanoparticles’ radiosensitizers evaluated in clinical trials for HNCs.

Collaborator	MNP Based Radiosensitizer	Study Phase	Mode of Administration	Age (yrs)	ClinicalTrials.gov Identifier (NCT #)	Study Start Date	Completion Date
Nanobiotix and M.D. Anderson Cancer Center Houston, Texas, United States	NBTXR3 (Hafnium oxide NPs)	1/2	Single intra-arterial injection	>65	NCT01946867	08/2013	08/2017
M.D. Anderson Cancer CenterHouston, Texas, United States	NBTXR3, Radiation Therapy, and Pembrolizumab	2	Intratumoral/ Intranodal	>18	NCT04862455	04/2021	09/2026 (Active)
National Institutes of Health Clinical Center (CC) National Cancer Institute (NCI)	CYT-6091 (TNF-bound colloidal gold)	1	Intravenous Administration	>18	NCT00356980	07/2006	03/2012

**Table 2 cancers-14-00514-t002:** MNPs alone and in combination with radiation have been explored as radiosensitizers to improve head and neck cancer directed therapy.

Nanoparticles	Size	Model Studied	Photon Radiation Dose/Energy
Gold nanoparticles (51)	1.9 nm	In vivo mouse head and neck squamous cell carcinoma model, SCCVII	42 Gy, 30 Gy, 50.6 Gy
Cetuximab-targeted gold nanoparticles (GNPs) (54)	30 nm	In vivo mouse A431 cells	25 Gy
Glucose and Cisplatin (CG-GNPs) (55)	20 nm	A431 cells for in vitro and in vivo mice experiments	6 MV
Gadolinium-based nanoparticles (GBNs) (60)	2.9 ± 0.2 nm	Radioresistant human head and neck squamous cell carcinoma (SQ20B, FaDu and Cal33 cell lines) and SQ20B tumor-bearing mouse model	10 Gy
Gadolinium-based nanoparticles (AGuIX^®^) (61)	5 nm	HNC cell lines (SQ20B, FaDu, and Cal33)	1–4 Gy
Gadolinium-based nanoparticles (AGuIX^®^) (62)	5 nm	Cal33 Orthotopic female NMRI nude mouse	10 Gy
Nanocomposite Ag/C225, constructed, which consisted of silver nanoparticles (AgNPs) conjugated to an epidermal growth factor receptor-specific antibody (C225) (65)	20 nm	Nasopharyngeal carcinoma epithelial (CNE)	6 MV X-ray irradiation (dose rate 200 cGy/min)

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
