# Peer review of "Therapeutic Advancements in Metal and Metal Oxide Nanoparticle-Based Radiosensitization for Head and Neck Cancer Therapy"

_cancers, 2022, doi:10.3390/cancers14030514_

Round 1

Reviewer 1 Report

It was a review study about utilizing metal (oxide) nanoparticles as photosensitizers for improving the treatment of head and neck cancer therapy. Here are some comments on this study which should be considered before publication:

  • There are some typo- and grammatically mistakes in the text which should be corrected.
  • "When combined with X-rays in vivo, head and neck tumor growth was inhibited. Because CYT-6091 has passed phase 1 trials (NCT00356980 and NCT00436410), this work is ripe for clinical translation []." Please add reference here.
  • You need to have better discussion about some of the samples you mentioned, for instance I here " In another one of the first studies examining NP in head and neck cancer, Teraoka et al., used HSC‑3 (human tongue squamous cell carcinoma) cells to show that X-ray irradiation was able to significantly reduce total cell number, and that the addition of AuNPs enhanced this suppressive effect on total cell number caused by X-ray irradiation. The reduction of total cell number by X-ray irradiation alone and when combined with AuNPs was attributed to the induction of apoptosis [45]." You need to mention the effect of Au on increasing the effect of X-ray irradiation in comparison to the other one, quantitatively.

In some cases (such as sample mention in section 4) you add a lot of data from one study that most of them are not necessary. Please mention just important data which are related to your study.

  • Please add more samples about utilizing gadolinium nanoparticles for radiation therapy. Also please mention the probable cytotoxicity of these nanoparticles on the normal cells.
  • "In summary, in the very first trial, the gadolinium-based AGuIX nanoparticles demonstrated the enhancement of radiotherapy after incomplete tumor resection in a HNC orthotropic animal model. Apart from metal nanoparticles, metal oxide nanoparticles have also been explored for their radiosensitization potential." This part is not good, please rewrite it.
  • Reference 58 didn't mention the effect of combination use of metal oxide nanoparticle and radiation therapy. Please replace it.
  • Heading "4" could be used as subheading of section "2".
  • Please mention the physicochemical features of nanoparticles which are needed for their application for head and neck cancer cells.

Reviewer 2 Report

In the paragraph “To address these challenges, during the last few decades there has been increasing interest in exploiting various nanomaterials (size 10-1000nm) including nanocarriers for drug delivery, metal nanoparticles (MNPs), drug nanoparticle conjugates, and nanocomposites, for cancer theranostics (therapy and diagnosis) application” The reviewer recommends to change some concepts: A metal nanoparticle can be used as nanocarriers for drug delivery and also a bimetal NPs is a nanocomposites… A description of few materials like nanoliposomes, nanomicelles, metallic nanoparticles… and later their potential applications like drug delivery and theranostics could be an option. In addition, this reference should be added:

DOI: 10.3390/pharmaceutics13030416

The sentence “MNPs passively diffuse into the tumor cells by a phenomenon referred to as enhanced permeation and retention effect (EPR).” needs some changes. The EPR effect is a passive tumor targeting, that is, the target are not tumor cells, is the tumor tissue.  Additionally, is more often named as Enhanced Permeability and Retention (EPR) effect. In addition, these references should be added:

DOI: 10.2174/1381612821666150820100812

The authors should check also lines 100 and 101 about the EPR.

Authors must check the sentence “Notably, it has been documented that while DNA is located in the cell nucleus, the nucleus is inaccessible even for nanoparticles of ultrafine dimensions (~2.5–10 nm)”, lines 104-105. There are papers demonstrating exactly the contrary

DOI: 10.1021/nn5008572

DOI: 10.1039/C9CC09728G

The paragraph: “Radiosensitization or dose augmentation produced by the addition of high Z atom MNPs  in target tumor tissues with photon radiation has been widely examined and recognized in the literature, while the use of MNPs with proton beam radiation continues to be an active area of research.” needs references.

Line 182, something is missing in the bracket.

Just mention about the figures, that they have low quality, in the final version please change them.

The paragraph “Iron oxide nanoparticles (SPIONS) are metal oxide nanoparticles that offer many exciting features that can be exploited in MRI imaging, in creating drug delivery systems and in enhancing cancer therapeutics.” needs references:

DOI: 10.1007/978-3-030-74073-3_3

DOI: 10.1021/acsami.1c02338

DOI: 10.1016/j.jcis.2020.06.011

DOI: 10.1021/acsami.8b18270

The reviewer recommends the inclusion of a table with all NPs, for instance:

Type of NPs / in vitro / in vivo (orthotopic / heterotopic) / dose …

Finally, overall, the reviewer recommends an actualization of references, related to Metallic NPS and radiotherapy. In 2021 the authors can find 340 publications about this topic.

For example, about Gold NPs as radiosensitizers

Author Response

Please see the attachedment.
